# The sharp structural switch of covalent cages mediated by subtle variation of directing groups

Qiong Chen[1,4], Zhaoyong Li[1,2,4], Ye Lei[1,4], Yixin Chen[1], Hua Tang[1], Guangcheng Wu [1], Bin Sun[3], Yuxi Wei[1], Tianyu Jiao[1], Songna Zhang[3] ✉, Feihe Huang [1,3] ✉, Linjun Wang [1,2] ✉ & Hao Li [1,3] ✉

It is considered a more formidable task to precisely control the self-assembled products containing purely covalent components, due to a lack of intrinsic templates such as transition metals to suppress entropy loss during self-assembly. Here, we attempt to tackle this challenge by using directing groups. That is, the self-assembly products of condensing a 1:2 mixture of a tetraformyl and a biamine can be precisely controlled by slightly changing the substituent groups in the aldehyde precursor. This is because different directing groups provide hydrogen bonds with different modes to the adjacent imine units, so that the building blocks are endowed with totally different conformations. Each conformation favors the formation of a specific product that is thus produced selectively, including chiral and achiral cages. These results of using a specific directing group to favor a target product pave the way for accomplishing atom economy in synthesizing purely covalent molecules without relying on toxic transition metal templates.

Mother Nature avoids byproducts in synthesis by taking advantage of reversible bonding forces[1-4]. The reversible nature of these supramolecular interactions allows synthetic errors to be checked and corrected. This biological capability inspires chemists to employ either noncovalent forces[5-7] or dynamic bonds[8-14] as the reaction motifs in synthesizing artificial systems. Henceforth, high-yielding syntheses could be accomplished without relying on tedious stepwise procedures, by sophisticatedly designing and tuning the geometries and conformations of the building blocks that allows the corresponding target products to represent the thermodynamic minima in self-assembly. One of the most successfully developed reversible reactions is metal-ligand coordination[15-28]. A variety of molecules with complex architectures and topologies have been successfully self-assembled[29-34], some of which[16,17,21-23,30] were obtained in close to quantitative yields. Here, transition metal cations with fixed coordination modes are able to dictate the corresponding organic ligands to

orientate in specific manners that favor the formation of some specific products. Subtle changes of the organic ligands in geometry and/or size might lead to dramatic variation in self-assembly pathway[35-42]. The same level of success, however, has not been achieved in the systems containing purely covalent components that are often relatively more flexible, in which the intrinsic templates namely transition metals are absent. The implication is that, the self-assembled products via dynamic covalent chemistry[8-14,43-45] such as imine formation[46-57] are thus often less controllable compared to the coordinative counterparts, despite a few exceptions[56,58-63]. For example, Cooper[64] et al. demonstrated that the amino precursors containing odd or even numbers of methylene units favored the formation of [2 + 3] or [4 + 6] cages, respectively. Mastalerz[13] et al. discovered that an organic cage self-assembled via boronic ester bond formation could undergo dimerization and form a catenane in solid-state, when switching the constitution of the side chains in the tetraol precursor. More recently,

[1]Department of Chemistry, Zhejiang University, Hangzhou 310058, PR China. [2]Key Laboratory of Excited-State Materials of Zhejiang Province, Zhejiang University, Hangzhou 310058, PR China. [3]ZJU-Hangzhou Global Scientific and Technological Innovation Center, Zhejiang University, Hangzhou 311215, PR China. [4]These authors contributed equally: Qiong Chen, Zhaoyong Li, Ye Lei. ✉e-mail: zsnchem@zju.edu.cn; fhuang@zju.edu.cn; ljwang@zju.edu.cn; lihao2015@zju.edu.cn

the same group[65] indicated that introducing methoxy or thiomethyl unit onto the framework of an imine cube led to occurrence of dimerization and trimerization, forming catenanes in solution. Mukherjee[66] et al. employed intramolecular hydrogen bonding to direct self-sorting. The group led by Beuerle[67] obtained a highly strained organic cage whose formation would be otherwise unlikely to occur without intramolecular driving forces namely hydrogen bonds.

In the present work, we employed directing groups to precisely control the self-assembly products based on imine formation. These directing groups mediate or determine the conformations of the building blocks, by providing hydrogen bonds with different modes to the latter. Each of these preorganized conformations of the building blocks favors one specific cage compound that is produced as the predominant product. To be more specific, a pseudo-linear tetraformyl precursor containing two isophthalaldehyde units and *trans*-cyclohexane-1,2-diamine (*trans*-CHDA), which is either enantiomerically pure or the racemic mixture, are combined for self-assembly. Subtle variation in the substituents located in each isophthalaldehyde residue switches the self-assembly products between two types of constitutionally different cage molecules, including a [3 + 6] chiral cage and a [2 + 4] achiral cage. In the case of OH unit whose acid proton acts as a hydrogen bond donor, intramolecular hydrogen bonds drive the two formyls and/or the resultant imine units on both sides to orientate in an *exo-endo* conformation. Such conformation favors the production of a [3 + 6] chiral cage, which is composed of three equivalents of the tetraformyl precursor and six equivalents of enantiomerically pure *trans*-CHDA. When the racemic *trans*-CHDA is used in self-assembly, narcissistic self-sorting occurs, generating a pair of enantiomers of the [3 + 6] chiral cage each containing only one type of enantiomer of *trans*-CHDA. As a comparison, an alkoxy substituent containing no acidic protons affords the two formyls and/or imines an

*exo-exo* conformation[68]. When the tetraformyl precursor is combined with the racemic mixture of *trans*-CHDA in a 1:2 ratio, a *meso* [2 + 4] cage is produced as the only observable product, which is composed of two equivalents of the tetraformyl precursor and four equivalents of racemic *trans*-CHDA. More interestingly, when the substituent is an ester unit containing protons with modest acidity, both *exo-endo* and *exo-exo* conformations become thermodynamically feasible. The self-assembled products are then determined by the chirality of the bisamine precursors. That is, enantiomerically pure and racemic bisamine favor the [3 + 6] and [2 + 4] products, respectively. Physicochemical analysis based on NMR spectroscopic results, solid-state structures, as well as theoretical calculation results indicated that these self-assembly preferences stem from the self-assembly products attempting to minimize intramolecular steric hindrance, by keeping all the imine protons in the *syn* conformation[69,70] relative to the corresponding adjacent methine proton in the cyclohexyl unit.

## Results

Each of the five structurally analogous tetraaldehyde precursors **0-4** (Fig. 1), whose synthetic procedures are described in the Supplementary Information (Supplementary Figs. 5, 7, 9, 11 and 14), contains two isophthalaldehyde units. The differences between these five precursors lie in the substituents grafted in each of the isophthalaldehyde moiety between the two formyl units (Fig. 1), which are –H (**0**), –OH (**1**), –OC$_4$H$_9$ (**2**), –OCH$_2$COOC$_2$H$_5$CH$_3$ (**3**), and –OCH$_2$COOC(CH$_3$)$_3$ (**4**), respectively. A pair of enantiomers of a chiral bisamine, namely (1*S*,2*S*)-cyclohexane-1,2-diamine ((*S*,*S*)-CHDA) and (1*R*,2*R*)-cyclohexane-1,2-diamine ((*R*,*R*)-CHDA), as well as ethylenediamine (EDA) were used as the amino partners for self-assembly (Fig. 1).

We first combined **1** (2.5 mM) and (*S*,*S*)-CHDA (Fig. 2A) in a 1:2 ratio in CDCl$_3$. After heating the solution at 50 °C for 6 h, the $^1$H NMR

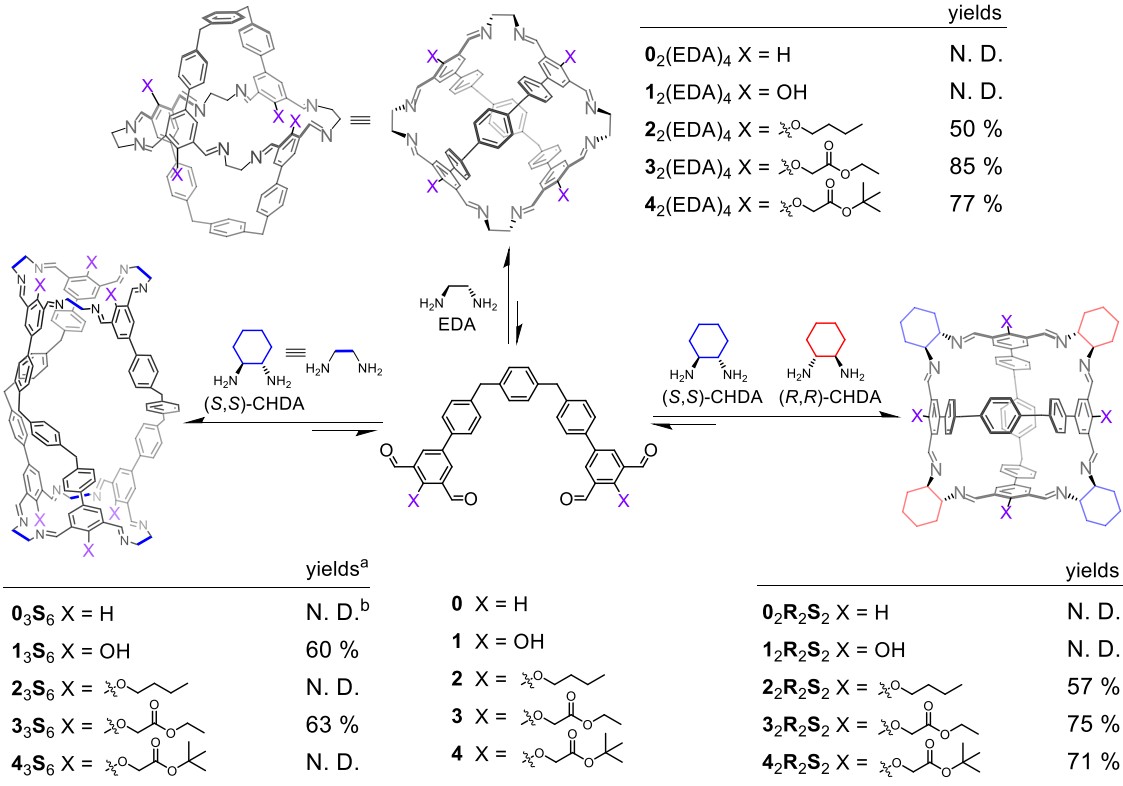

| | yields |
|---|---|
| **0₂(EDA)₄** X = H | N. D. |
| **1₂(EDA)₄** X = OH | N. D. |
| **2₂(EDA)₄** X = ⤳O⟍⟍ | 50 % |
| **3₂(EDA)₄** X = ⤳O⟍COO⟍ | 85 % |
| **4₂(EDA)₄** X = ⤳O⟍COO⟍ | 77 % |

| | yields[a] | | | | yields |
|---|---|---|---|---|---|
| **0₃S₆** X = H | N. D.[b] | **0** X = H | | **0₂R₂S₂** X = H | N. D. |
| **1₃S₆** X = OH | 60 % | **1** X = OH | | **1₂R₂S₂** X = OH | N. D. |
| **2₃S₆** X = ⤳O⟍⟍ | N. D. | **2** X = ⤳O⟍⟍ | | **2₂R₂S₂** X = ⤳O⟍⟍ | 57 % |
| **3₃S₆** X = ⤳O⟍COO⟍ | 63 % | **3** X = ⤳O⟍COO⟍ | | **3₂R₂S₂** X = ⤳O⟍COO⟍ | 75 % |
| **4₃S₆** X = ⤳O⟍COO⟍ | N. D. | **4** X = ⤳O⟍COO⟍ | | **4₂R₂S₂** X = ⤳O⟍COO⟍ | 71 % |

**Fig. 1 | Structural formulae and NMR yields of a series of cage products including 2₂R₂S₂, 3₂R₂S₂, 4₂R₂S₂, 0₃S₆, 1₃S₆, 3₃S₆, 2₂(EDA)₄, 3₂(EDA)₄, 4₂(EDA)₄.** These cages are produced by condensing each of the corresponding tetraformyl precursors including **0-4**, and the bisamino partner namely racemic *trans*-CHDA, (*S*,*S*)-CHDA, or EDA. [a]The yields are all determined by using internal standard in the corresponding $^1$H NMR samples without isolating the corresponding products. [b]N. D. means Not Determined, because either the target molecules were not produced with observable yields, or the products are generated within a library of mixture whose $^1$H NMR spectrum is too complicated to determine the corresponding yields.

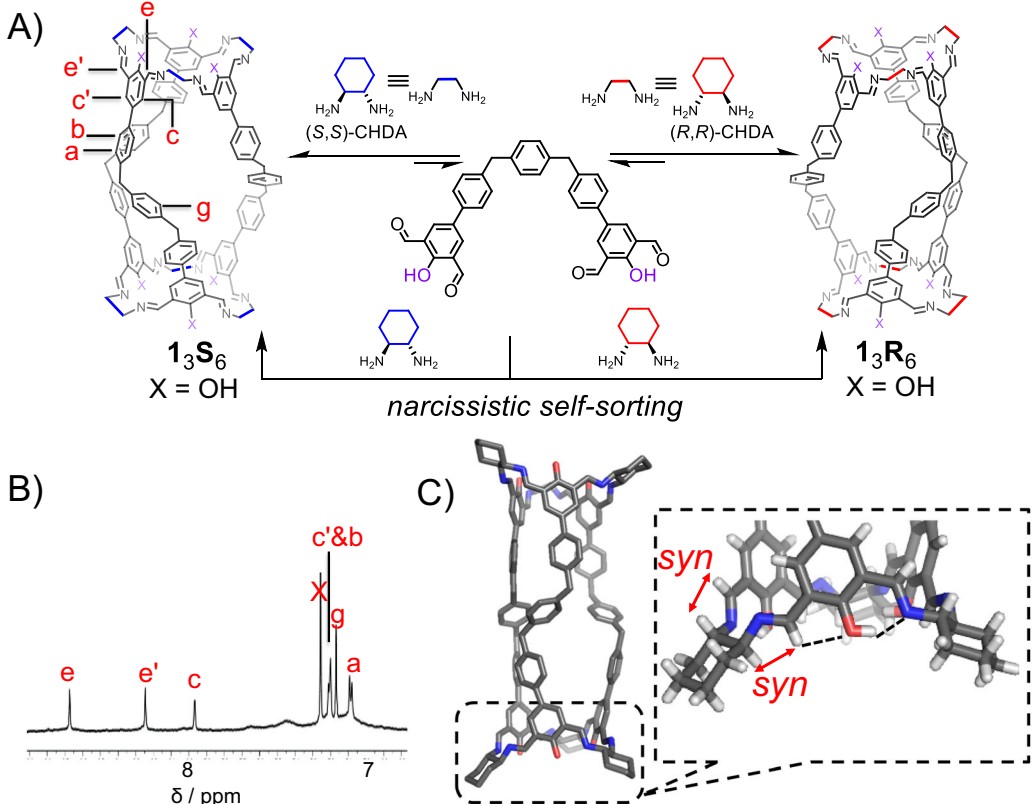

**Fig. 2 | The self-assembly of a pair of cage enantiomers $1_3S_6$ and $1_3R_6$.**
(**A**) Structural formulae of $1_3S_6$ and $1_3R_6$ by condensing the precursor **1** and the corresponding bisamino partners namely either (*S,S*)-CHDA or (*R,R*)-CHDA. Narcissistic self-sorting occurs when **1** is combined with the racemic mixture of (*S,S*)-CHDA and (*R,R*)-CHDA. (**B**) Partial $^1$H NMR spectrum (600 MHz, CDCl$_3$, 298 K) of $1_3S_6$. Some of the key resonances are labeled, which were assigned based on the corresponding two-dimensional NMR spectra shown in Supplementary Figs. 31 and 32. The full spectrum is also shown in Supplementary Fig. 29. (**C**) Solid-state

structures of $1_3S_6$ obtained from single-crystal X-ray diffraction analysis. Oxygen atoms, red; nitrogen, blue; carbon, gray. Hydrogen atoms and disordered solvent molecules are omitted for clarity. We also expanded the terminal part of the cage $1_3S_6$ for the sake of clarity. It is clearly observed that all the imine protons adopt the *syn* conformation relative to the corresponding methine protons, as labeled with red double-head arrows. Two intramolecular hydrogen bonds between the central OH proton and the imine nitrogen on one side, and the central OH oxygen and the imine proton on the other side, are labeled with black dashed lines.

spectrum (Fig. 2B) was recorded, in which a set of sharp resonances were observed, indicating a product with a symmetrical structure was obtained as the predominant product. Mass spectrum (Supplementary Fig. 28) indicated that this product is a [3 + 6] product, namely composed of three equivalents of **1** and six equivalents of (*S,S*)-CHDA. This product is referred to as $1_3S_6$. In the $^1$H NMR spectrum (Fig. 2B), each of the resonances corresponding to the protons in the isophthalaldehyde residues, including both the imine *e/e'* and the phenyl *c/c'*, splits into two peaks. This observation indicated that in each isophthalaldehyde residue, the two imine units have two different orientations, namely either *exo* or *endo* with respect to the central OH group (Fig. 3B, middle). The framework of $1_3S_6$ is chiral, resulting from the stereo chirality of the (*S,S*)-CHDA precursor. The chirality of $1_3S_6$ was supported by $^1$H NMR spectrum (Supplementary Fig. 29) in which the two protons in the methylene *f* become diastereotopic. The solid-state sample of $1_3S_6$ was obtained by adding MeOH into its solution in chloroform and collecting the precipitate via filtration. The solid of $1_3S_6$ was re-dissolved in CDCl$_3$, whose $^1$H NMR spectrum was essentially the same as the before precipitation, indicated that $1_3S_6$ was rather kinetically inert and did not undergo observable degradation during precipitation and re-dissolving. In the $^1$H NMR spectrum (Fig. 2B) of $1_3S_6$, a few small broad resonances were observed, indicating that some oligomeric or polymeric byproducts were also generated. These broad resonances were not removed, even after we attempted to purify $1_3S_6$ via precipitation. We thus used NMR yield to quantify the production of each cage in this article, by adding an internal standard

in the NMR sample. The self-assembly yield of $1_3S_6$ was determined to be 60% (Supplementary Fig. 76). The cage $1_3R_6$, which is the enantiomer of $1_3S_6$, was also self-assembled in a similar procedure (Supplementary Fig. 16C), by condensing **1** and (*R,R*)-CHDA in CDCl$_3$. The circular dichroism (CD) spectra (Supplementary Fig. 33B) of both $1_3S_6$ and $1_3R_6$ were recorded, showing mirror-like images. When **1** was combined with a racemic mixture of (*R,R*)-CHDA and (*S,S*)-CHDA in CDCl$_3$, narcissistic self-sorting[71–73] occurred (Supplementary Fig. 17), yielding a racemic mixture of $1_3S_6$ and $1_3R_6$ as the major product.

Single crystals of the cage $1_3S_6$ (Fig. 2C), as well as the racemic mixture (Supplementary Fig. 88) of $1_3S_6$ and $1_3R_6$, were obtained by vapor diffusion of methanol or acetonitrile into the corresponding solutions in CHCl$_3$ or DMF, respectively. The solid-state structures of both $1_3S_6$ and $1_3R_6$ convinced the *exo-endo* conformation of the two imine units in each isophthalaldehyde residue, consistent with the $^1$H NMR spectroscopic results. This *exo-endo* conformation is favored by the formation of intramolecular hydrogen bonds (Fig. 3B, middle), namely that the oxygen atom in the central OH forms hydrogen bond with the imine proton on one side, while the OH proton forms hydrogen bond with the imine nitrogen atom on the other side. These two types of hydrogen bonds are clearly observed in the solid-state structures (Fig. 2C, black dashed lines). Such *exo-endo* conformation was also observed in many cage systems containing *trans*-CHDA[61,74–76]. The *exo-endo* conformation allows all the imine protons to orientate in a *syn* conformation relative to the corresponding adjacent methine protons (Fig. 2C, red double-head arrows). According to a report by

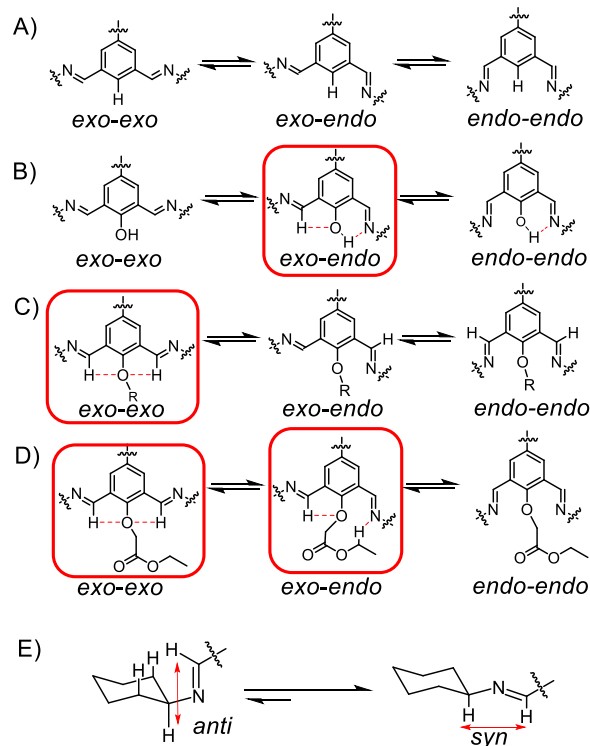

**Fig. 3 | The possible conformations of the imine adducts of isophthalaldehyde derivatives bearing different central substituents between the two imine units.** These different central substituents include (**A**) H, (**B**) OH, (**C**) alkoxy (OR) unit, and (**D**) ester (OCH₂COOC₂H₅) chain. Hydrogen bonding interactions occurs between the central substituents and the two imine units on both sides, which are marked with red dashed lines. OH favors the *exo-endo* conformation, while the OR unit favors *exo-exo* one. In the case of the ester, both *exo-exo* and *exo-endo* are stable conformations, driven by hydrogen bonding with different modes. These relatively stable conformations are encircled with red rectangles. (**E**) Two conformations of an imine compound, including *syn* (right) and *anti* (left). The *syn* conformer is more favored than the *anti* one due to smaller steric hindrance in the former conformer.

Gawronski[70], such *syn* conformer is thermodynamically more favored than the *anti* counterpart by 0.92 kcal/mol, due to larger steric hindrance in the latter conformer (Fig. 3E). We also combined **1** and EDA, which is a less preorganized counterpart of *trans*-CHDA. Heating the mixture in CDCl₃ led to the generation of a library of oligomeric and polymeric precipitates, instead of a putative [3 + 6] cage namely **1₃(EDA)₆**. Such results indicate that conformation preorganization is also of importance in the bisamino precursors.

In order to strengthen our proposition that the OH group in **1** plays a predominant role in favoring the formation of the corresponding [3 + 6] cages namely either **1₃S₆** or **1₃R₆**, we synthesized three analogs including **0, 2** as well as **3**, in which the central OH substituents in **1** are replaced by other units namely H, alkyloxy and ester respectively. The precursor **0** can be considered as a counterpart of **1** without any directing groups. **0** and (*S,S*)-CHDA were combined in CDCl₃ in a 1:2 ratio. After heating the corresponding solution at 50 °C for 4 h, the ¹H NMR spectrum (Supplementary Fig. 23A) indicated that an analog of **1₃S₆**, namely **0₃S₆** was produced as the major product, accompanied with a [2 + 4] product **0₂S₄** as a kinetic product. The formation of both **0₃S₆** and **0₂S₄** was convinced by mass spectrometry (Supplementary Fig. 24). Further heating the reaction mixture for 120 h converted most **0₂S₄** into **0₃S₆**, as inferred from the observation that the molecular ion peak corresponding to the [2 + 4] product weakened significantly (Supplementary Fig. 25). Even heating the solution for no less than 120 h, the resonances corresponding to **0₂S₄** with small intensity were still observable in the ¹H NMR spectrum (Supplementary Fig. 23C),

indicating that the conversion from **0₂S₄** into **0₃S₆** was not complete. The NMR yield of **0₃S₆** was determined to be 70% (Supplementary Fig. 84). Compared to **1**, the precursor **0** is lack of the OH directing group, so that the **0₃S₆** is thermodynamically less favored. Combining **0** and the racemic CHDA in a 1:2 ratio also produced the racemic mixture of **0₃S₆** and **0₃R₆** as major products. However, this narcissistic self-sorting was much less successful compared to the aforementioned system involving **1**, i.e., more oligomeric or polymeric byproducts (Supplementary Fig. 27) were observed in the corresponding ¹H NMR spectrum.

We then combined either **2** or **3** (2.5 mM) with a racemic mixture of *trans*-CHDA in a 1:2 ratio in CDCl₃. After heating at 50 °C for 6 h, the ¹H NMR spectra (Fig. 4B and Supplementary Fig. 48) and mass spectra (Supplementary Figs. 34 and 47) of both solutions were recorded. Mass spectrum indicated that in both cases, [2 + 4] products were generated. That is, each product is composed of two equivalents of tetraformyl precursors (i.e., **2** or **3**) and four equivalents of *trans*-CHDA, namely either (*S,S*)-CHDA or (*R,R*)-CHDA.

Single crystals of these two self-assembled products were obtained by vapor diffusion of either methanol or diethyl ether into the corresponding solutions in CHCl₃, respectively. The solid-state structures unambiguously indicated that two cage products, namely **2₂R₂S₂** and **3₂R₂S₂** were obtained (Fig. 4C–F). In the framework of both **2₂R₂S₂** and **3₂R₂S₂**, the two imine bonds in each of the isophthalaldehyde residues orientate in an *exo-exo* manner. This conformation is in sharp contrast to the *exo-endo* conformation in the case of either **1₃S₆** or **1₃R₆** containing OH substituents. The *exo-exo* conformation is favored by the formation of intramolecular five-member ring hydrogen bonds between the central oxygen atom and the two imine protons on both sides (Fig. 3C, left), assisted by the repulsion between the central oxygen and two imine nitrogen atoms. In the framework of either **2₂R₂S₂** or **3₂R₂S₂**, the four *trans*-CHDA residues distribute in an *RSRS* manner. That is, each isophthalaldehyde residue is connected by two different enantiomers of *trans*-CHDA. Such ligand distribution affords each cage namely either **2₂R₂S₂** or **3₂R₂S₂** two plane symmetries, affording both **2₂R₂S₂** and **3₂R₂S₂** *meso* structures. In addition, as expected, each imine proton adopts the more favored *syn* conformation with respect to the corresponding axial methine proton.

The ¹H NMR spectra (Fig. 4B and Supplementary Fig. 48) of both **2₂R₂S₂** and **3₂R₂S₂** are similar. In the ¹H NMR spectrum (Fig. 4B) of **2₂R₂S₂**, a few singlets corresponding to the protons including *e, c* and *g* are observed. The simple patterns of both ¹H NMR spectra convince that both **2₂R₂S₂** and **3₂R₂S₂** are highly symmetrical, consistent with the corresponding solid-state structure. It is noteworthy that because both (*R,R*)-CHDA and (*S,S*)-CHDA are involved in self-assembly, the cage containing four *trans*-CHDA residues has theoretically nine stereoisomers. However, the ¹H NMR spectra unambiguously indicates that only the *RSRS* ones, namely **2₂R₂S₂** or **3₂R₂S₂**, is produced selectively. This is because the *RSRS* isomer is the only one that can allow all the imine protons to adopt the more favored *syn* conformation with respect to corresponding axial methine protons. This unlikely occurring social self-sorting behavior was also reported by Mastalerz recently[77]. Both **2₂R₂S₂** and **3₂R₂S₂** were isolated as solid-state compounds by adding MeOH into the corresponding solutions in chloroform and collecting the precipitates via filtration. Again, NMR yields were used to quantify their production, given that oligomeric or polymeric impurities rendered the isolated yields less accurate. By using internal standard in the corresponding NMR samples, the yields (Supplementary Figs. 77 and 79) of **2₂R₂S₂** and **3₂R₂S₂** were determined to be 57% and 75%, respectively. We also combined either **2** or **3** (2.5 mM) with EDA (5.0 mM) in CDCl₃, yielding two self-assembled products whose ¹H NMR spectra (Supplementary Figs. 43 and 53) are very similar as those of **2₂R₂S₂** and **3₂R₂S₂**. Mass spectra (Supplementary Figs. 42 and 52) confirmed that two cages namely **2₂(EDA)₄** and

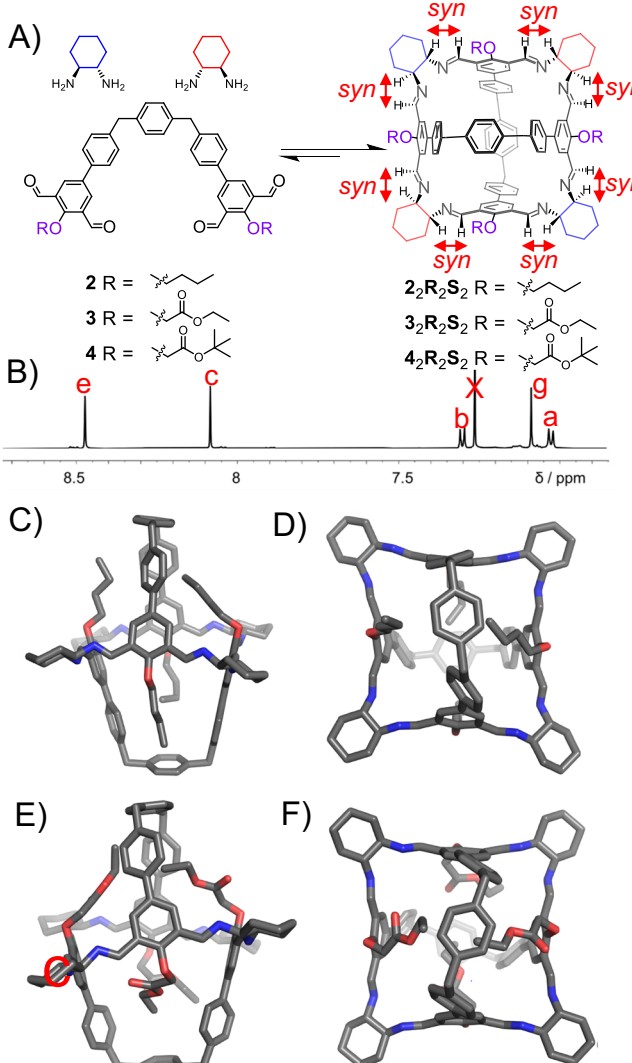

**Fig. 4 | The self-assembly of a series of achiral cages, including $2_2R_2S_2$, $3_2R_2S_2$ and $4_2R_2S_2$. (A)** Structural formulae of the cages $2_2R_2S_2$, $3_2R_2S_2$ and $4_2R_2S_2$, by condensing a 2:1 mixture of racemic *trans*-CHDA and the corresponding tetra-formyl precursor namely **2**, **3** and **4**, respectively. **(B)** Partial ¹H NMR spectrum (600 MHz, CDCl₃, 298 K) of $2_2R_2S_2$. Some of the key resonances are labeled, which were assigned based on the corresponding two-dimensional NMR spectra shown in the Supplementary Figs. 37 and 38. The full spectrum is shown in the Supplementary Fig. 35. Solid-state structures of $2_2R_2S_2$ including **(C)** side view and **(D)** top view, as well as $3_2R_2S_2$ including **(E)** side view and **(F)** top view, which were obtained from single-crystal X-ray diffraction analysis. Oxygen atoms, red; nitrogen, blue; carbon, gray. Hydrogen atoms and disordered solvent molecules are omitted for clarity.

$3_2(EDA)_4$ were self-assembled, whose yields (Supplementary Figs. 78 and 80) are 50 % and 85 %, respectively. It is still unclear why the yields of $3_2R_2S_2$ and $3_2(EDA)_4$ are higher compared to $2_2R_2S_2$ and $2_2(EDA)_4$. We hypothesize that the precursor **3** contains two ester side chains. The protons in the methylene units that are connected directly by the ester oxygen atoms are more acidic compared to the methylene protons in **2**. The former protons form stronger CH-π interactions with the phenyl units in the cage framework.

When the racemic *trans*-CHDA was replaced by enantiomeric pure (S,S)-CHDA, self-assembly yielded totally different products. Heating a 1:2 mixture of **2** (2.5 mM) and (S,S)-CHDA in CDCl₃ produced a library of mixture, whose ¹H NMR spectrum (Supplementary Fig. 19B) showed a set of irregular resonances. Such observation indicated that the

putative chiral cage namely $2_2S_4$ is not a thermodynamically favored product. The putative cage $2_2S_4$ can be considered as a chiral counterpart of $2_2R_2S_2$, in which two (R,R)-CHDA residues are replaced by two (S,S)-CHDA residues. However, in $2_2S_4$, four of the eight imine protons adopt the thermodynamically disfavored *anti* conformation (Supplementary Fig. 19) with respect to the adjacent methine protons, making its formation less favored. Another putative product, namely $2_3S_6$, an analog of $1_3S_6$, was not observed in either ¹H NMR spectrum or mass spectrum. This is not surprising, given that the [3 + 6] products require the *exo-endo* conformation, while the alkyloxy directing groups in **2**, favor the *exo-exo* conformation in contrast.

To our surprise, combining the tetraaldehyde **3** (2.5 mM) and (S,S)-CHDA in CDCl₃ in a 1:2 ratio yielded a product whose ¹HNMR spectrum (Fig. 5B) was in reminiscence of the aforementioned [3 + 6] product namely $1_3S_6$. Mass spectrum (Supplementary Fig. 57) confirms that a chiral cage $3_3S_6$ was self-assembled as the predominant product, whose yield was determined to be 63 % (Supplementary Fig. 81). As occurred in $1_3S_6$, the resonances corresponding to two imine protons in $3_3S_6$ also split into two peaks (Fig. 5B), indicating that the two imines within each isophthalaldehyde residue adopt an *exo-endo* conformation. The *exo-endo* conformation was also supported by the NOESY spectrum (Supplementary Fig. 61) of $3_3S_6$. That is, while the *endo* imine proton *e'* undergoes coupling with one phenyl proton *c'* in the isophthalaldehyde residue, the *exo* imine proton *e* undergoes coupling with the CH₂ proton *k* in the central side chain. The *exo-endo* conformation in the case of $3_3S_6$ was also driven by the formation of two different types of hydrogen bonds (Fig. 3D, middle). That is, one hydrogen bond forms between *exo* imine proton *e* and the central oxygen atom, while the second one forms between the *endo* imine nitrogen atom and the proton in the methylene unit *h/h'* in ethyl unit of the ester. The occurrence of the latter hydrogen bonding results from the electron-withdrawing inductive effect of the ester oxygen atom, which renders the protons *h/h'* relatively acidic. Such relatively acidic protons are absent in the case of **2** containing *n*-butoxy chains. It is therefore not surprising that **2** did not form the putative cage $2_3S_6$. The occurrence of CH–N hydrogen bonding forces in $3_3S_6$ is supported by its ¹HNMR spectrum (Fig. 5B), in which the resonances corresponding to the methylene unit *h/h'* in the ester split into two peaks, while the methylene unit *k* exhibits a singlet. In order to further strengthen our hypothesis that the acidic ester proton plays an important role in cage formation, we thus synthesized another tetraformyl precursor **4** (see its molecular formula in Fig. 1) containing *tert*-butyl acetate units. **4** contains ester functions while does not contain the protons with modest acidity as those in **3**. Combining **4** and (S,S)-CHDA in a 1:2 ratio yielded a library of mixture (Supplementary Fig. 22), instead of $4_3S_6$. This control experiment unambiguously supported our proposition that hydrogen bonding involving the ester side chains in **3** plays a critical role in favoring the formation of $3_3S_6$. Addition of (R,R)-CHDA into $3_3S_6$ gradually transformed latter into the [2 + 4] achiral cage $3_2R_2S_2$ (Fig. 5A). Furthermore, combining a mixture of pre-synthesized $3_3S_6$ and $3_3R_6$ with a 1:1 ratio also yielded the meso cage $3_2R_2S_2$. Apparently, the achiral cage $3_2R_2S_2$ is more favored in terms of entropy compared with $3_3S_6$, because the latter is composed of fewer building blocks compared to the former.

In order to confirm that the self-assembly preference results from the cage products attempting to keep all the imine protons in the *syn* conformation with respect to the corresponding methine protons, we further performed density functional theory (DFT) calculations at the BP86-D3/6-311G(d) level with the Gaussian 16 package[78]. Based on the solid-state structure of $2_2R_2S_2$ obtained via crystallography, two putative cages, namely $2_2S_4$ and $2_2S_2R_2$, were optimized (see details in Supplementary Data 1). Here, $2_2S_4$ is a putative counterpart of $2_2R_2S_2$, whose two (R,R)-CHDA residues are replaced with two (S,S)-CHDA residues. $2_2S_2R_2$ was obtained by replacing the two (R,R)-CHDA and two (S,S)-CHDA residues with (S,S)-CHDA and (R,R)-CHDA,

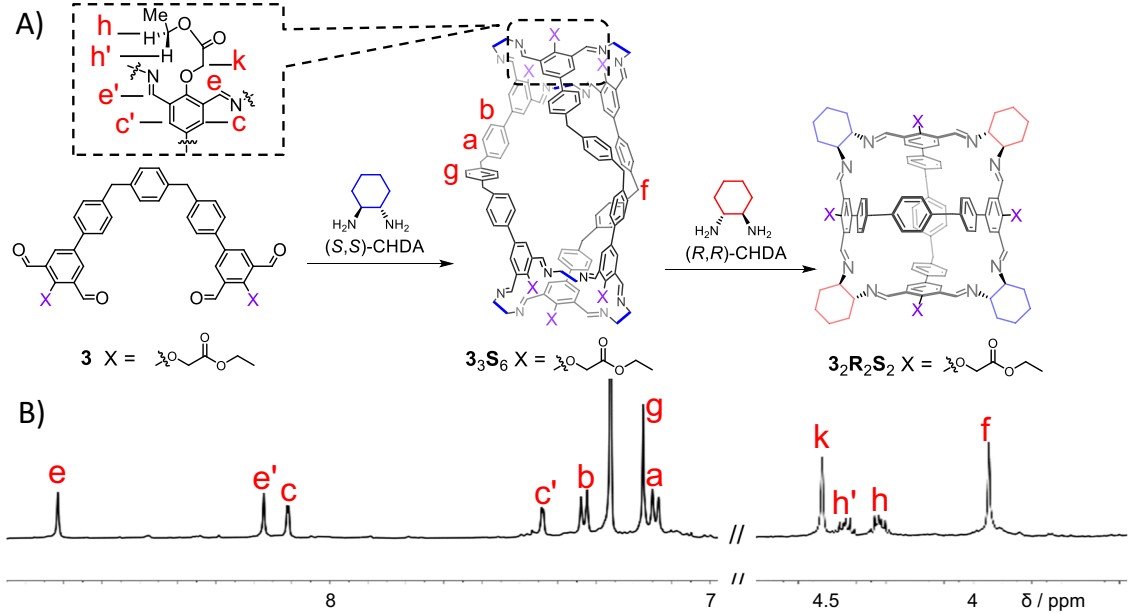

**Fig. 5 | The self-assembly of a chiral cage $3_3S_6$ that was transformed into $3_2R_2S_2$ upon addition of (R,R)-CHDA.** (**A**) Structural formula of the cage $3_3S_6$, by condensing a 2:1 mixture of (S,S)-CHDA and **3**. Upon addition of (R,R)-CHDA, $3_3S_6$ is converted into $3_2R_2S_2$. (**B**) Partial [1]H NMR spectrum (600 MHz, CDCl$_3$, 298 K) of $3_3S_6$. Some of the key resonances are labeled, which were assigned based on the corresponding two-dimensional NMR spectra shown in the Supplementary Figs. 60 and 61. The full spectrum is shown in the Supplementary Fig. 58.

respectively. In $2_2R_2S_2$, $2_2S_4$ and $2_2S_2R_2$, there are respectively zero, four and eight imine protons that adopt the less favored *anti* conformation with respect to the corresponding methine protons. The theoretical results revealed that the free energies of $2_2S_4$ and $2_2S_2R_2$ are 5.4 kcal/mol and 9.1 kcal/mol with respect to that of $2_2R_2S_2$, respectively (Fig. 6). This confirms that the *syn* conformer is thermodynamically more favored than the *anti* counterpart in the cage framework. $2_2R_2S_2$, with all the imine protons in the *syn* conformation, is the most stable and favored product, which is fully consistent with our experimental results. Similar approaches were also used to calculate the free energies of $1_3S_6$ and a putative cage $1_3R_6(anti)$ (Supplementary Fig. 89). It is noteworthy that $1_3S_6$ and $1_3R_6(anti)$ are not a pair of enantiomers. The putative cage $1_3R_6(anti)$ was obtained by replacing all (S,S)-CHDA residues in $1_3S_6$ with (R,R)-CHDA, while keeping the conformations of all the imine bonds (details see Supplementary Data 1). In the putative cage $1_3R_6(anti)$, all imine protons adopt the *anti* conformation with respect to the adjacent methine protons. The calculations revealed that the free energy of $1_3R_6(anti)$ is 37.4 kcal/mol higher than that of $1_3S_6$ (Supplementary Fig. 89).

## Discussion

In summary, an ingenious approach to precisely control the self-assembly product based on imine condensation is developed, by introducing different directing groups to favor specific target products. When a tetraformyl precursor containing two isophthalaldehyde units and *trans*-CHDA are combined, the self-assembly products are very sensitive to the variation of the central substituents located in each isophthalaldehyde residue between two imine units. This is because the central directing groups provide hydrogen bonding with different modes to imine building blocks located on both sides. These intramolecular forces endow the imine units with specific conformations, each resembling and favoring a specific cage product that is produced selectively. To be specific, in the case of OH substituent group whose proton is rather acidic, the two imine bonds are pre-organized in an *exo-endo* conformation, driven by two different hydrogen bonds in the form of either CH–O or OH–N. Such conformation favors the formation of a chiral [3 + 6] cage containing

enantiomeric pure *trans*-CHDA residues. When the substituent group contains no acidic protons such as an alkoxyl unit, the two imine bonds have an *exo-exo* conformation driven by two identical hydrogen bonds namely CH–O. Such conformation favors the formation of achiral [2 + 4] cages containing racemic *trans*-CHDA building blocks as the predominant product, implying the occurrence of social self-sorting. In the case of ester substituent group containing protons with modest acidity, both the *exo-endo* and *exo-exo* conformations are thermodynamically feasible. As a consequence, the self-assembly products are surprisingly determined by the chirality of the *trans*-CHDA, namely that enantiomeric pure and racemic *trans*-CHDA favor the [3 + 6] and [2 + 4] products respectively. Such preference results from a tendency that the imine protons attempt to adopt the *syn* conformation relative to the adjacent axial methine protons in the CHDA residues, in order to minimize steric hindrance. Our fundamental understanding of precisely controlling the thermodynamic stability of a target product by using directing group to regulate the intramolecular forces, is thus significantly improved. Such results help us to rule out the reliance of cationic transition metal templates that might be highly toxic. Future research includes self-assembly of cage molecules with larger cavities and water-solubility, so that these self-assembled hosts are employed for applications in some more challenging arenas, such as mimicking enzyme[79–82] in artificial systems.

## Methods

### Self-assembly of $1_3S_6$

A 1:2 mixture of **1** (2.77 mg, 0.005 mmol) and (S,S)-CHDA (1.14 mg, 0.01 mmol) was combined and dissolved in CDCl$_3$ (2 mL). The corresponding reaction mixture was heated at 50 °C for 6 h. $1_3S_6$ was self-assembled as the major product in the corresponding [1]H NMR spectrum, without further manipulation. The solid-state sample of $1_3S_6$ was obtained by adding MeOH into its solution in chloroform and collecting the precipitate via filtration. However, it is unsuccessful to purify $1_3S_6$ via precipitation, accompanied by the generation of some oligomeric or polymeric byproducts. We thus used a NMR yield (60 %) to quantify the production of $1_3S_6$, by adding an internal standard in the NMR sample.

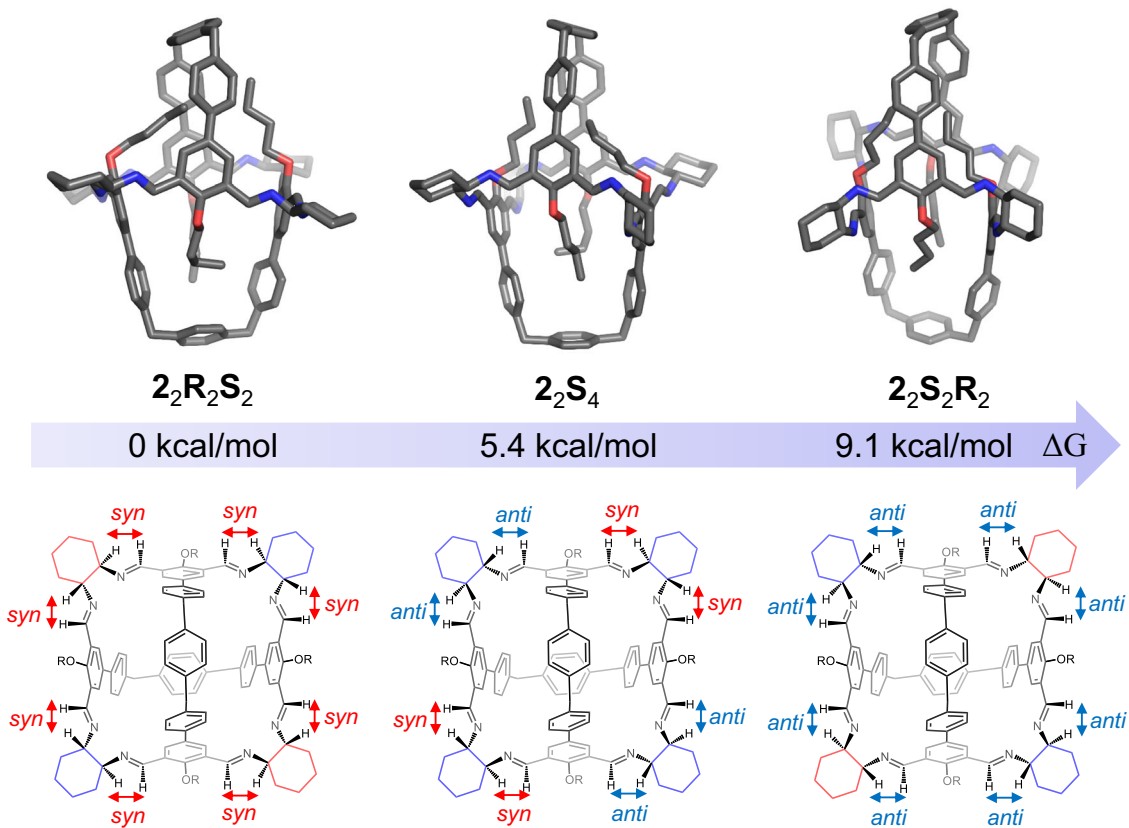

**Fig. 6 | The optimized structures (top) and the structural formula (bottom) of $2_2R_2S_2$ and its two putative less stable counterparts $2_2S_4$ and $2_2S_2R_2$, with their corresponding relative free energies.** The calculation was performed by using DFT calculations at the level of BP86-D3 functional and 6-311 G(d) basis set.

### Self-assembly of $2_2R_2S_2$, $3_2R_2S_2$ and $4_2R_2S_2$

A 1:2 mixture of the corresponding tetraformyl precursor (**2**, **3** or **4**, 0.005 mmol) and racemic *trans*-CHDA (1.14 mg, 0.01 mmol) was combined and dissolved in CDCl₃ (2 mL). The corresponding reaction mixture was heated at 50 °C for 6 h. A [2 + 4] achiral cage ($2_2R_2S_2$, $3_2R_2S_2$ or $4_2R_2S_2$) was self-assembled as the major product in the ¹H NMR spectrum, without further manipulation. The solid-state sample of each cage was obtained by adding MeOH into its solution in chloroform and collecting the precipitate via filtration. However, it is also unsuccessful to purify these cages via precipitation. We thus used NMR yields to quantify the production of $2_2R_2S_2$ (57 %), $3_2R_2S_2$ (75 %) and $4_2R_2S_2$ (71 %), by adding an internal standard in the NMR sample.

### Self-assembly of $2_2(EDA)_4$, $3_2(EDA)_4$ and $4_2(EDA)_4$

A 1:2 mixture of the corresponding tetraformyl precursor (**2**, **3** or **4**, 0.005 mmol) and EDA (0.60 mg, 0.01 mmol) was combined and dissolved in CDCl₃ (2 mL). The corresponding reaction mixture was heated at 50 °C for 6 h. A [2 + 4] achiral cage ($2_2(EDA)_4$, $3_2(EDA)_4$ or $4_2(EDA)_4$) was self-assembled as the major product in the ¹H NMR spectrum, without further manipulation. The NMR yield of $2_2(EDA)_4$, $3_2(EDA)_4$ and $4_2(EDA)_4$ was determined to be 50%, 85% and 77%, respectively.

### Self-assembly of $3_3S_6$

A 1:2 mixture of **3** (3.63 mg, 0.005 mmol) and (*S,S*)-CHDA (1.14 mg, 0.01 mmol) was combined and dissolved in CDCl₃ (2 mL). The corresponding reaction mixture was heated at 50 °C for 12 h. $3_3S_6$ was self-assembled as the major product in the corresponding ¹H NMR spectrum, without further manipulation. The NMR yield of $3_3S_6$ was determined to be 63%.

### Self-assembly of $0_3S_6$

A 1:2 mixture of **0** (3.13 mg, 0.006 mmol) and (*S,S*)-CHDA (1.37 mg, 0.012 mmol) was combined and dissolved in CDCl₃ (2 mL). The corresponding reaction mixture was heated at 50 °C for 24 h. $0_3S_6$ was self-assembled as the major product accompanied with a [2 + 4] product as a kinetic product in the corresponding ¹H NMR spectrum. As the reaction proceeding, the [2 + 4] kinetic product would transfer to the [3 + 6] chiral cage mostly, but not completely. The NMR yield of $0_3S_6$ was determined to be 70%.

### General methods

Nuclear magnetic resonance (NMR) spectra were recorded at ambient temperature using Bruker AVANCE III 400, Bruker AVANCE III 500, or Agilent DD2 600 spectrometers, with working frequencies of 400/500/600 and 100/125/150 MHz for ¹H and ¹³C, respectively. Chemical shifts are reported in ppm relative to the residual internal non deuterated solvent signals (CDCl₃: $\delta$ = 7.26 ppm, DMSO-$d_6$: $\delta$ = 2.50 ppm). High-resolution mass spectra (HRMS) were measured by using a SHIMADZU liquid chromatograph mass spectrometry ion trap time of flight (LCMS-IT-TOF) instrument and Bruker Daltonics Autoflex III (MALDI-TOF). X-ray crystallographic data were collected on a Bruker D8 Venture diffractometer. CD spectra were recorded on a Circular Dichroism Spectrometer (Chirascan V100, Applied Photophysics Ltd).

### Theoretical calculations

All investigated cage structures were optimized by using the density functional theory (DFT) at the BP86-D3/6-311G(d) level with the Gaussian 16 package[78]. The solvent effect of chloroform was included with the polarizable continuum model (PCM)[83]. All the optimized structures were verified by the phonon frequencies calculated at the same level (namely no imaginary frequency should exist).

## Data availability

The authors declare that all other data supporting the findings of this study are available from the article and its Supplementary Information. Cartesian coordinates and energies of all investigated cage structures are provided in Supplementary Data 1. The X-ray crystallographic coordinates for structures reported in this study have been deposited at the Cambridge Crystallographic Data Centre (CCDC), under deposition numbers 2201823, 2201825, 2203208 and 2203209. These data can be obtained free of charge from The Cambridge Crystallographic Data Centre via www.ccdc.cam.ac.uk/structures. Additional data are available from the authors upon request.

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

## Acknowledgements

The research at Zhejiang University was supported by the Starry Night Science Fund of Zhejiang University Shanghai Institute for Advanced Study (No. SN-ZJU-SIAS-006 to H.L.). H.L. also want to thank the

support from the Leading Innovation Team grant from Department of Science and Technology of Zhejiang Province (2022R01005). L.W. acknowledges support from the National Natural Science Foundation of China (No. 22273082) and the High Performance Computing Center in Department of Chemistry, Zhejiang University.

## Author contributions

H.L. conceived the project. Q.C., Z.L., Y.L., Y.C., S.Z., F.H., L.W. and H.L. prepared the manuscript. Q.C., Y.L., H.T., B.S. and Y.W. synthesized the molecules studied in this work. Q.C., Y.C., G.W., T.J. and S.Z. performed characterization of the key compounds. Q.C. and H.L analyzed experimental data and draw conclusions. Under the supervision of L.W., Z.L. performed theoretical calculations and analysis. All authors discussed the results and commented on the manuscript.

## Competing interests

The authors declare no competing interests.
