## [Peer Review File · Nature Communications]

REVIEWER COMMENTS

Reviewer #1 (Remarks to the Author):

The manuscript entitled "The Sharp Structural Switch of Covalent Cages Mediated by Subtle Variation of Directing Groups" describes the usage of directing groups to precisely control the self-assembly products via imine formation. The directing groups can preorganize the two formyl/imine bonds on both sides into specific conformations. The corresponding product that has this conformation would become thermodynamically more favored, and therefore can be self-assembled selectively as the predominant product, including chiral [3+6] cages and [2+4] achiral cages. Even although using directing groups to control the self-assembly products is not rare, sharp structural switch by slightly modulating the structures of directing groups is still not easy. In addition, the achiral [2+4] cage is composed of two equivalents of tetraformyl and four equivalents of racemic bisamine. Its self-assembly thus implies the occurrence social self-sorting involving three precursors, which is a nontrivial process, given that only one isomer among nine possible products is generated selectively. Another interesting discovery is that, the ester functional group in aldehyde precursor has a switchable directing ability, when different bisamino precursors are used. That is, the directing behavior of the ester unit is similar as alkoxy unit, generating a [2+4] cage when racemic bisamino precursor is used. However, when enantiomerically pure bisamine is used, a chiral [3+6] cage was formed, by taking advantage of its weakly acidic ester CH proton. In fact, the acidity of CH protons is sometimes overlooked in the field of supramolecular chemistry. Based on these discoveries, I recommend its publication in Nature Communications after minor revision.

1. In figure 2B, is there a hydrogen bond in the endo-endo form between the OH proton and imine nitrogen?

2. In figure 5, the authors demonstrated that combining the precursor 2 and enantiomerically pure (S,S)-CHDA yielded cage products with "open" architectures containing unreacted formyls or amino units. Even although ¹H NMR spectroscopy and mass spectrometry are used to characterize these molecules, the evidence of their formation is still less convincing, given that these products are not isolated and only characterized within the library of mixture. Is it possible to purify these molecules?

3. The authors mainly focus on the studies of self-assembly preference. The authors should at least discuss the possibility of using these cages for host-guest chemistry and other applications.

4. The authors emphasize the importance of the acidity of ester protons. Is it possible to further increase or alter the yield by using amide instead of ester? or adding a cyano or CF₃ group on this carbon?

Reviewer #2 (Remarks to the Author):

Chen et al. reported the controllable assembly and self-sorting behavior of imine-based organic cages by H-bonding via adjusting the substituent groups in the aldehyde precursor. In addition to the examples provided by the authors, there have been many reports on controllable assembly and self-sorting (J. Am. Chem. Soc. 2017, 139, 50, 18142–18145 ; Nat Commun 2016, 7, 12469 ; J. Am. Chem. Soc. 2022, 144, 14, 6180–618 ; Angew. Chem. Int. Ed. 2019, 58, 16275). Moreover, the intramolecular H-bonding direct imine-based cage assembly has been reported (Chem. Eur. J. 2014, 20, 1646-1657, Chem. Eur. J. 2021, 27, 6077-6085). Considering Nature Communications is as a high-level general journal, the novelty of this work is not high enough for publishing on Nature Communications. Hence, I am afraid that I am unable to recommend acceptance.

The following issues need the authors' attention.

(1) In CDCl₃, the authors discuss the assembly of aldehyde amine monomers extensively, e.g. they claim that " No peaks corresponding to 22S4 were observed, further supporting our hypothesis that in

the presence of enantiomeric pure bisamine, the final ring closure reaction of [22S4 + H₂O] to form 22S4 is thermodynamically disfavored." This is only the result in chloroform. It is well known that solvation effects have a great influence on the course of chemical reactions, apparently, the authors missed this point. Other solvent wasn't considered, such as toluene, DMSO, DMF, chlorobenzene, acetonitrile, and ethyl acetate. Why? (Cryst. Growth Des. 2018, 18, 5, 2759–2764).

(2) To illuminate the self-assembly mechanism, the author conducted extensive NMR spectroscopy and cited theoretical data in literature. It is recommended that the author conduct theoretical calculations for their own systems to further study the self-assembly mechanism.

(3) For new compounds, the authors lack sufficient characterization support such as IR, elemental analysis, thermogravimetric analysis. PXRD patterns should be provided for crystalline samples. BET data is also important for porous structure.

(4) There are a large number of Class B alarms in the author's crystallography file, please check and resolve them, or explain them item by item in the cif file and generate a new CHECKCIF file.

Although the authors are presenting their views at the level of single-molecule assembly, a description of the crystal framework based on single-crystal data is still necessary.

Reviewer #3 (Remarks to the Author):

This manuscript reports a piece of interesting self-assembly result on cage formation via dynamic chemistry namely imine, in which subtle changes in the tetraaldehyde precursors induced dramatic changes in cage product structures, whose formation stoichiometries are switched between [3+6] and [2+4]. Indeed, the precise construction of organic cages with well-defined stoichiometry, topology and size is still a huge challenge in the field. The authors present a very elegant concept that using a substituent between the two formyls in an isophthalaldehyde unit can fix the relative orientations of the adjacent aldehyde/imine units. The products whose building blocks have such orientations are thus self-assembled as the major products. This approach might be even extended to other dynamic covalent architectures with complex topologies such as knots and interlocked molecules. I recommend its publication in Nature Communications after revision.

1) Some rather complex NMR spectra revealed that the precursor 0 (with H as the substituent) and CHDA yielded ill-defined mixtures of oligomers instead of cages with well-defined structures. However, the complicated ¹H NMR spectra can also be attributed to the production of a mixture of many cage diastereomers. It is likely that, exo-exo, exo-endo, endo-endo orientations can co-exist, since H is unable to fix the orientations of imine. Maybe the authors should provide mass spectra to rule out this possibility. 2) The authors demonstrated in Figure 6 that the precursor 3 with ester substituents forms a chiral cage when reacting with enantiomerically pure CHDA. Addition of the other CHDA enantiomer transformed the chiral [3+6] cage into a smaller achiral [2+4] one. What will happen, if the two [3+6] cage enantiomers are pre-formed first and then mixed? Will the two chiral cages kinetically trapped or be transformed into the more stable achiral one?

**The Sharp Structural Switch of Covalent Cages
Mediated by Subtle Variation of Directing Groups**

*Letter of Response to the
Reviewers' Comments*

Contents:

- Response to Comments from Reviewer # 1
- Response to Comments from Reviewer # 2
- Response to Comments from Reviewer # 3

Reviewer 1:

Comment: The manuscript entitled “The Sharp Structural Switch of Covalent Cages Mediated by Subtle Variation of Directing Groups” describes the usage of directing groups to precisely control the self-assembly products via imine formation. The directing groups can preorganize the two formyl/imine bonds on both sides into specific conformations. The corresponding product that has this conformation would become thermodynamically more favored, and therefore can be self-assembled selectively as the predominant product, including chiral [3+6] cages and [2+4] achiral cages. Even although using directing groups to control the self-assembly products is not rare, sharp structural switch by slightly modulating the structures of directing groups is still not easy. In addition, the achiral [2+4] cage is composed of two equivalents of tetraformyl and four equivalents of racemic bisamine. Its self-assembly thus implies the occurrence social self-sorting involving three precursors, which is a nontrivial process, given that only one isomer among nine possible products is generated selectively. Another interesting discovery is that, the ester functional group in aldehyde precursor has a switchable directing ability, when different bisamino precursors are used. That is, the directing behavior of the ester unit is similar as alkoxy unit, generating a [2+4] cage when racemic bisamino precursor is used. However, when enantiomerically pure bisamine is used, a chiral [3+6] cage was formed, by taking advantage of its weakly acidic ester CH proton. In fact, the acidity of CH protons is sometimes overlooked in the field of supramolecular chemistry. Based on these discoveries, I recommend its publication in Nature Communications after minor revision.

Response: We thank the reviewer for reading our manuscript so carefully and summarize the significance of our paper clearly. We also thank the reviewer for the positive comments.

Comment: In figure 2B, is there a hydrogen bond in the *endo-endo* form between the OH proton and imine nitrogen?

Response: In the *endo-endo* form, a hydrogen bond does form between the OH proton and one of the two imine nitrogens. We modified the Figure 2B to make this point clearer. It is noteworthy that since the OH proton can only form one hydrogen bond, the *endo-endo* conformation is less stable than the *exo-endo* one, in which two hydrogen bonds form simultaneously.

Comment: In figure 5, the authors demonstrated that combining the precursor **2** and enantiomerically pure (S,S)-CHDA yielded cage products with “open” architectures containing unreacted formyls or amino units. Even although ¹H NMR spectroscopy and mass spectrometry are used to characterize these molecules, the evidence of their formation is still less convincing, given that these products are not isolated and only characterized within the library of mixture. Is it possible to purify these molecules?

Response: We agree with the reviewer that using just NMR and mass spectra to characterize **2₂S₃** or **2₂S₅** is not fully convincing. In fact, **2₂S₃** or **2₂S₅** with “open” architectures bearing unreacted formyl or amino unit are not as stable as the “closed” counterparts in which all formyl and amino units form imine bonds. It is therefore our attempts to isolate these two products were unsuccessful.

We thus synthesized an aldehyde precursor **S1**, containing only three formyl units. **S1** is an analogue of **2**, in which one formyl is missing, and two of the phenyl units are replaced with pyridinium for the sake of easier synthesis. Condensing three equivalents of (S,S)-CHDA and two equivalents of precursor **S1** produced **S2** in 92% yield. **S2** was fully characterized via ¹H NMR spectroscopy and mass spectrometry, and single-crystal X-ray diffraction analysis. The ¹H NMR spectrum of **S2** is similar as that of **2₂S₃**. The assignment of each proton in **S2** in the NMR spectrum was done based on the two-dimensional NMR spectroscopic results. To our surprise, the solid-state structure of **S2** implies that our prediction of the

structures of either 2_2S_3 or 2_2S_5 are totally wrong. In **S2**, all the imine protons adopt *syn* conformation with respect to the cyclohexane methine protons. We thus rewrote the manuscript and SI. The mistaken discussion of either 2_2S_3 or 2_2S_5 were deleted.

Because we are writing another paper based on **S2**, we chose to not add its information in either this manuscript or the SI.

Comment: The authors mainly focus on the studies of self-assembly preference. The authors should at least discuss the possibility of using these cages for host-guest chemistry and other applications.

Response: We thank the reviewer for raising this point. We did try to use these cages for host-guest chemistry in organic phase. In $CDCl_3$, no host-guest chemistry was observed. The [3+6] cages, e.g., 1_3R_6 , have no cavities available to accommodate guests, given that their phenyl units undergo collapse with each other, which was clearly observed in the solid-state structure. In the case of [2+4] cages, the substituent groups, such as alkoxy in $2_2R_2S_2$ and ester units in $3_2R_2S_2$, insert into the cage cavities. This is also observed in the corresponding solid-state structures. As a consequence, most of part of the cages are occupied and no longer available for guest recognition. It is noteworthy that, the interactions between the side chains and the cage frameworks in fact act as the driving forces to drive the cage formation. For example, changing the alkoxy units in $2_2R_2S_2$ into MeO unit made the cage formation less successful, i.e., a variety of oligomeric and polymeric byproducts were also observed. In order to realize host-guest

recognition, future work would include, 1) using shorter side chain or enlarging the volume of the cage cavity, to guarantee that the cage cavity is not occupied, 2) introducing water-soluble units into the cage so that the cage can employ hydrophobic effect to drive host-guest recognition. Such trials are ongoing in our group.

In this paper, what we focus on is not host-guest chemistry. In fact, we focus on unravelling the underneath mechanism how the cages become the most thermodynamically favored one among the self-assembly library and how they can be precisely constructed without generating other byproducts.

Comment: The authors emphasize the importance of the acidity of ester protons. Is it possible to further increase or alter the yield by using amide instead of ester? or adding a cyano or CF₃ group on this carbon?

Response: We thank the reviewer for raising this point. As requested, we attempted to replace electron-withdrawing groups to further increase the acidity of the ester protons. Cyano and CF₃ are good electron-withdrawing units. However, the corresponding ester precursors containing either cyano or CF₃ units are not commercially available.

not commercially available

Then we turned our attention to amide, in which the NH proton is also rather acidic. However, the synthesis is not unsuccessful either, due to solubility issues. We synthesized a series of linear rigid tetraformyl precursors, **S3**, **S4**, **S5** and **S6**, bearing OH, ester, amide, and OBU chains, respectively. We combined each of **S3**, **S4**, **S5** and **S6**, and (S,S)-CHDA in CDCl₃. In the case of **S3**, **S4**, **S5**, the corresponding [3+6] cages were self-assembled as the predominant products. While in the case of **S6** bearing alkoxy units without acidic protons, intractable mixture was obtained. These results act as indirect evidence, convincing the importance of acidic protons in the self-assembly of [3+6] cages including **3**:**S6**.

It is noteworthy that we have already performed a control experiment in the original version of manuscript. That is, combining the tetraformyl precursor **4** containing tert-butyl acetate units and (S,S)-CHDA did not produce the cage product as **3** (Figure S12). Both **3** and **4** contain ester side chains, while **4** does not bear acidic protons. Such experiments unambiguously proved that the acidic protons in **3** play a predominant role in templating the formation of **3₃S₆**.

Reviewer 2:

Comment: Chen et al. reported the controllable assembly and self-sorting behavior of imine-based organic cages by H-bonding via adjusting the substituent groups in the aldehyde precursor. In addition to the examples provided by the authors, there have been many reports on controllable assembly and self-sorting (J. Am. Chem. Soc. 2017, 139, 50, 18142–18145; Nat Commun 2016, 7, 12469; J. Am. Chem. Soc. 2022, 144, 14, 6180–618; Angew. Chem. Int. Ed. 2019, 58, 16275). Moreover, the intramolecular H-bonding direct imine-based cage assembly has been reported (Chem. Eur. J. 2014, 20, 1646-1657, Chem. Eur. J. 2021, 27, 6077-6085).

Response: We thank the reviewer for providing a number of important examples of controllable self-assembly assembly and self-sorting of imine-based cages. In fact, a few of them have already been cited in our original version of manuscript. The others have been added into the revised version of the manuscript.

Comment: Considering Nature Communications is as a high-level general journal, the novelty of this work is not high enough for publishing on Nature Communications. Hence, I am afraid that I am unable to recommend acceptance.

Response: We agree with the reviewer in the statement that the research results in this manuscript was done by obtaining inspiration from others. However, we would like to make a humble rebuttal here that, taking advantage of subtle variation to switch self-assembly product is not as trivial as the reviewer thinks. Many discoveries in this manuscript represents the first one in the community of supramolecular chemistry. First, we realized both narcissistic and social self-sorting simultaneously in our system. That is, in the cage containing OH units, racemic bisamine would generate racemic [3+6] cages each containing only one enantiomer of the bisamine. In contrast, each the meso [2+4] cages contains both enantiomers of the bisamine, is produced selectively as the only product among the nine possible isomers. Second, we observed that the CH proton in the ester side chain is acidic enough to act as a hydrogen bond donor, so that the ester side chain can template both the [2+4] and [3+6] cages by using amine with different chirality. We believe that these novelties will be recognized in the community of supramolecular chemistry and dynamic covalent chemistry.

Comment: The following issues need the authors' attention.

In CDCl₃, the authors discuss the assembly of aldehyde amine monomers extensively, e.g. they claim that "No peaks corresponding to **2₂S₄** were observed, further supporting our hypothesis that in the presence of enantiomeric pure bisamine, the final ring closure reaction of [**2₂S₄** + H₂O] to form **2₂S₄** is thermodynamically disfavored." This is only the result in chloroform. It is well known that solvation effects have a great influence on the course of chemical reactions, apparently, the authors missed this point. Other solvent wasn't considered, such as toluene, DMSO, DMF, chlorobenzene, acetonitrile, and ethyl acetate. Why? (Cryst. Growth Des. 2018, 18, 5, 2759–2764).

Response: First, in the past weeks, we did many experiments, try to obtain the single crystal structures of **2₂S₃**, [**2₂S₄** + H₂O] and **2₂S₅**, all of which bearing unreacted formyl or amino groups. However, these compounds are not as stable as their "closed" counterparts. We just synthesized **S1**, an analogue of **2**. The former bears one less formyl unit compared to the latter precursor. Combining **S1** and CHDA produced **S2**, which were fully and unambiguously characterized via NMR, mass and crystallography. **2₂S₃** should have a similar structure as that of **S2**, except that the two unreacted formyl units in the former are removed in **S2**. The proposed structures of **2₂S₃**, [**2₂S₄** + H₂O] and **2₂S₅** in the former version of

manuscript are totally mistaken. We rewrote the manuscript and modified all the mistakes in the modified version of the manuscript.

We also thank the reviewer for raising this point that solvent might have great impact on the self-assembly outcomes. As requested, we tried self-assembly in all of the solvents suggested by the reviewer. We have experimentally confirmed that chloroform is the most suitable solvent. In MeCN, DMSO, DMF or ethyl acetate, combining **2** and (S,S)-CHDA in a 1:2 ratio led to precipitation, making self-assembly unsuccessful. For example, the ¹H NMR spectra recorded in MeCN and DMSO-*d*₆ were shown below (Figure A and B), indicating that after precipitation, the self-assembly solution became very dilute. In toluene or chlorobenzene, the self-assembly is similar as chloroform (Figure C and D), and no solvent effects were observed in these two cases.

^1H NMR spectra (400 MHz, CDCl_3 , 298 K) of products by condensing **2** and (*S,S*)-CHDA in a 1:2 ratio at 50 °C in different solvents, including A) CD_3CN , B) $\text{DMSO}-d_6$, C) toluene- d_8 and D) chlorobenzene- d_5 .

Comment: To illuminate the self-assembly mechanism, the author conducted extensive NMR spectroscopy and cited theoretical data in literature. It is recommended that the author conduct theoretical calculations for their own systems to further study the self-assembly mechanism.

Response: We thank the reviewer for providing such a suggestion. As requested, we asked Professor Linjun Wang, one of theoretical chemists, to perform theoretical calculation for these cages. These results are totally consistent with our experimental ones. We modified the main-text (Figure 6) and SI (Figure S79), introducing these results.

Optimized structures (top) and structural formula (bottom) of different [2+4] analogues, namely $2_2\text{R}_2\text{S}_2$, 2_2S_4 and $2_2\text{S}_2\text{R}_2$, with the relative free energies.

All cage structures were optimized by using the density functional theory (DFT) at the BP86-D3/6-311G(d) level with the Gaussian 16 package. Based on the structure of $2_2R_2S_2$, we replaced two (*R,R*)-CHDA residues with two (*S,S*)-CHDA residues and got a putative cage 2_2S_4 , in which four of the eight imine protons adopt the *anti* conformation with respect to the adjacent methine protons. By a similar way, we replaced all CHDA residues of $2_2R_2S_2$ with different configurational CHDA residues and got a putative cage $2_2S_2R_2$, in which all of the eight imine protons adopt the *anti* conformation. The calculation performed by using DFT revealed that the relative free energies of $2_2R_2S_2$, 2_2S_4 and $2_2R_2S_2$ is 0 kcal/mol, 5.4 kcal/mol and 9.1 kcal/mol, respectively. This result strongly confirms that *syn* conformer is thermodynamically more favored than the *anti* counterpart. As a consequent, $2_2R_2S_2$, with all the imine protons in the *syn* conformation, is the most stable and favored product, which is completely consistent with our experiment results.

Optimized structures of different [3+6] cages, namely 1_3S_6 and $1_3R_6(anti)$, with the relative free energies.

Similar approaches were also used to calculate the free energies of 1_3S_6 and a putative cage $1_3R_6(anti)$. It is noteworthy that 1_3S_6 and " $1_3R_6(anti)$ " described here are not a pair of enantiomers. The putative cage $1_3R_6(anti)$ was obtained by replacing all (*S,S*)-CHDA residues in 1_3S_6 with (*R,R*)-CHDA residues, while keeping the conformations of all the imine bonds. In the putative cage $1_3R_6(anti)$, all imine protons adopt the *anti* conformation with respect to the adjacent methine protons. The calculation performed by using DFT revealed that the relative free energies of 1_3S_6 and $1_3R_6(anti)$ is 0 kcal/mol and 37.4 kcal/mol, respectively. This result also strongly confirms that *syn* conformer is thermodynamically more favored than the *anti* counterpart, which is completely consistent with our experiment results.

Comment: For new compounds, the authors lack sufficient characterization support such as IR, elemental analysis, thermogravimetric analysis. PXRD patterns should be provided for crystalline samples. BET data is also important for porous structure.

Response: We thank the reviewer for providing such a suggestion. As requested, we added IR and elemental analysis to the aldehyde precursors in SI.

In addition, we attached IR, thermogravimetric analysis and PXRD pattern of cage $2_2R_2S_2$ in SI (Figures S29-31) and below. It is noteworthy that, NMR spectroscopy, mass spectrometry and single-crystal X-ray

diffraction represent more reliable and modern techniques to unambiguously characterize our cages. We thus did not perform IR, thermogravimetric analysis and PXRD to other cage products. In addition, BET data is often used to characterize the porous materials for gas absorption properties. In our systems, we focus on discussing how to precisely construct self-assembled cages, instead of host-guest recognition properties. We thus it is less important to obtain the BET data.

FT-IR spectrum of $2_2R_2S_2$.

PXRD pattern of $2_2R_2S_2$.

TGA of $2.2R_2S_2$ solid.

Comment: There are a large number of Class B alarms in the author's crystallography file, please check and resolve them, or explain them item by item in the cif file and generate a new CHECKCIF file.

Response: As requested, we have checked and resolved the Class B alarms item by item in cif files. In addition, we have generated new CHECKCIF files.

Comment: Although the authors are presenting their views at the level of single-molecule assembly, a description of the crystal framework based on single-crystal data is still necessary.

Response: As requested, we added the description of the crystal framework in the SI (Figures S75-78).

Reviewer 3:

Comment: This manuscript reports a piece of interesting self-assembly result on cage formation via dynamic chemistry namely imine, in which subtle changes in the tetraaldehyde precursors induced dramatic changes in cage product structures, whose formation stoichiometries are switched between [3+6] and [2+4]. Indeed, the precise construction of organic cages with well-defined stoichiometry, topology and size is still a huge challenge in the field. The authors present a very elegant concept that using a substituent between the two formyls in an isophthalaldehyde unit can fix the relative orientations of the adjacent aldehyde/imine units. The products whose building blocks have such orientations are thus self-assembled as the major products. This approach might be even extended to other dynamic covalent architectures with complex topologies such as knots and interlocked molecules. I recommend its publication in Nature Communications after revision.

Response: We thank the reviewer for reading our manuscript so carefully and summarize the significance of our paper clearly. We also thank the reviewer for the positive comments.

Comment: Some rather complex NMR spectra revealed that the precursor **0** (with H as the substituent) and CHDA yielded ill-defined mixtures of oligomers instead of cages with well-defined structures. However, the complicated ¹H NMR spectra can also be attributed to the production of a mixture of many cage diastereomers. It is likely that, exo-exo, exo-endo, endo-endo orientations can co-exist, since H is unable to fix the orientations of imine. Maybe the authors should provide mass spectra to rule out this possibility.

Response: We thank the reviewer for bringing this issue to our attention. We did repeat the self-assembly of the precursor **0** and CHDA. We extended the reaction time of self-assembly, and we realized that longer reaction time led to more successful self-assembly in the case of **0**. Our previous observation that combining **0** and CHDA only produced oligomeric or polymeric byproducts might result from the fact that the system did not reach equilibrium. Given long enough reaction time, the ¹H NMR spectra are relatively clean (see Figures below). A [3+6] chiral cage is self-assembled as the major product by condensing precursor **0** and (S,S)-CHDA, accompanied with a [2+4] product as a minor kinetic product. As the reaction proceeding, the amount of [2+4] kinetic product would transfer into the thermodynamic [3+6] chiral cage. However, the transformation was not complete, i.e., small amount of [2+4] product was still observed in the ¹H NMR spectrum even after the system reached the equilibrium. Both [2+4] and [3+6] products are confirmed via mass spectrometry. In the case of **0** which lacks of directing groups to do preorganization, longer reaction time is required for the system to reach the final product. When the precursor **0** and racemic CHDA are combined in a 1:2 ratio, [3+6] chiral cages are also the major product. However, the self-assembly was less successful compared to enantiomerically pure bisamine, i.e., more oligomeric and polymeric byproducts were generated and observed.

Based on these results, we modified the maintext and SI (Figures S13-17) accordingly.

Partial ¹H NMR spectra (400 MHz, CDCl₃, 298 K) of products by condensing **0** (3 mM) and (S,S)-CHDA in a 1:2 ratio at 50 °C for different time, including A) 4 h, B) 24 h and C) 120 h. The resonances of [3+6] cage and [2+4] product are marked with blue and red arrows, respectively.

DOSY spectrum of the mixed self-assembly products by condensing **0** and (S,S)-CHDA at 50 °C for 5 h. .

Mass spectra of the self-assembly product by condensing **.0** (3 mM) and (S,S)-CHDA at 50 °C for 4 h.

Mass spectra of the self-assembly product by condensing **.0** (3 mM) and (S,S)-CHDA at 50 °C for 120 h.

Partial ^1H NMR spectrum (400 MHz, CDCl_3 , 298 K) of products by condensing **0** and racemic *trans*-CHDA in a 1:2 ratio at 50°C for 24 h. The resonances of [3+6] cages are marked with blue arrows.

Comment: The authors demonstrated in Figure 6 that the precursor **3** with ester substituents forms a chiral cage when reacting with enantiomerically pure CHDA. Addition of the other CHDA enantiomer transformed the chiral [3+6] cage into a smaller achiral [2+4] one. What will happen, if the two [3+6] cage enantiomers are pre-formed first and then mixed? Will the two chiral cages kinetically trapped or be transformed into the more stable achiral one?

Response: If two [3+6] cage enantiomers, namely **3₃R₆** and **3₃S₆**, are pre-formed first and then mixed, they would be totally transformed into the more stable achiral cage, namely **3₂R₂S₂**. This experiment result is strongly confirmed by the corresponding ^1H NMR spectra (400 MHz, CDCl_3 , 298 K) below. This result indicates that the achiral cage **3₂R₂S₂** is more favored in terms of entropy compared with the chiral cage **3₃R₆** or **3₃S₆**, because the latter is composed of fewer building blocks compared to the former.

Partial ^1H NMR spectra (400 MHz, $CDCl_3$, 298 K) of A) 3_3S_6 , B) 3_3R_6 and C) the 1:1 reaction mixture of 3_3S_6 and 3_3R_6 .

REVIEWERS' COMMENTS

Reviewer #1 (Remarks to the Author):

The authors have addressed properly all issues I raised. Several new building blocks were prepared and tested for self-assembly. Failed attempts were well explained. To identify open-form species, the authors synthesised a compound with three formyl units that cannot form a cage, Instead, a pseudo[1]catanene was found, which indicated previously predicted structures were wrong. The authors have corrected those parts in the manuscript and SI. I do not have further comments, and thus would recommend to accept this manuscript.

Reviewer #2 (Remarks to the Author):

It is clear from their response that Li and coworkers have carefully considered the reviewers' comments, which generally pertained to the below key points:

The novelty

Missing supporting information from experimental and theoretical aspects

Insufficient literature references

The revisions, which are accounted for in the response document, have addressed these key points. Following these changes, the manuscript could be published as it.

Reviewer #3 (Remarks to the Author):

I suggest to accept the manuscript without revision.

**The Sharp Structural Switch of Covalent Cages
Mediated by Subtle Variation of Directing Groups**

*Letter of Response to the
Reviewers' Comments*

Contents:

- Response to Comments from Reviewer # 1
- Response to Comments from Reviewer # 2
- Response to Comments from Reviewer # 3

Reviewer #1 (Remarks to the Author):

Comment:

The authors have addressed properly all issues I raised. Several new building blocks were prepared and tested for self-assembly. Failed attempts were well explained. To identify open-form species, the authors synthesised a compound with three formyl units that cannot form a cage. Instead, a pseudo[1]catanene was found, which indicated previously predicted structures were wrong. The authors have corrected those parts in the manuscript and SI. I do not have further comments, and thus would recommend to accept this manuscript.

Response: We thank the reviewer for providing a relatively positive comment and allowing publication of this paper in *Nat. Commun.*

Reviewer #2 (Remarks to the Author):

Comment:

It is clear from their response that Li and coworkers have carefully considered the reviewers' comments, which generally pertained to the below key points:

The novelty

Missing supporting information from experimental and theoretical aspects

Insufficient literature references

The revisions, which are accounted for in the response document, have addressed these key points. Following these changes, the manuscript could be published as it.

Response: We thank the reviewer for providing a relatively positive comment and allowing publication of this paper in *Nat. Commun.*

Reviewer #3 (Remarks to the Author):

Comment:

I suggest to accept the manuscript without revision.

.

Response: We thank the reviewer for providing a relatively positive comment and allowing publication of this paper in *Nat. Commun.*